# Maternal Iron Deficiency Programs Rat Offspring Hypertension in Relation to Renin—Angiotensin System and Oxidative Stress

**DOI:** 10.3390/ijms23158294

**Published:** 2022-07-27

**Authors:** Ya-Hui Chang, Wan-Hsuan Chen, Chung-Hao Su, Hong-Ren Yu, You-Lin Tain, Li-Tung Huang, Jiunn-Ming Sheen

**Affiliations:** 1Department of Pediatrics, Chiayi Chang Gung Memorial Hospital and Chang Gung University College of Medicine, Chiayi 61344, Taiwan; dcvforgood@gmail.com (Y.-H.C.); wererabbit122@hotmail.com (W.-H.C.); b9302049@cgmh.org.tw (C.-H.S.); 2Department of Pediatrics, Kaohsiung Chang Gung Memorial Hospital and Chang Gung University College of Medicine, Kaohsiung 83301, Taiwan; yuu2004taiwan@yahoo.com.tw (H.-R.Y.); tainyl@hotmail.com (Y.-L.T.); litung.huang@gmail.com (L.-T.H.)

**Keywords:** hypertension, iron deficiency, oxidative stress, programming, renin-angiotensin system

## Abstract

Hypertension is an important public health challenge, affecting up to 30–50% of adults worldwide. Several epidemiological studies indicate that high blood pressure originates in fetal life—the so-called programming effect or developmental origin of hypertension. Iron-deficiency anemia has become one of the most prevalent nutritional problems globally. Previous animal experiments have shown that prenatal iron-deficiency anemia adversely affects offspring hypertension. However, the underlying mechanism remains unclear. We used a maternal low-iron diet Sprague Dawley rat model to study changes in blood pressure, the renal renin-angiotensin system, oxidative stress, inflammation, and sodium transporters in adult male offspring. Our study revealed that 16-week-old male offspring born to mothers with low dietary iron throughout pregnancy and the lactation period had (1) higher blood pressure, (2) increased renal cortex angiotensin II receptor type 1 and angiotensin-converting enzyme abundance, (3) decreased renal cortex angiotensin II receptor type 2 and MAS abundance, and (4) increased renal 8-hydroxy-2′-deoxyguanosine and interleukin-6 abundance. Improving the iron status of pregnant mothers could influence the development of hypertension in their offspring.

## 1. Introduction

Hypertension is an important public health challenge, affecting approximately 30–50% of adults worldwide [1]. Hypertension is associated with a considerably higher risk of adverse cardiovascular and renal outcomes including heart failure, ischemic stroke, intracerebral hemorrhage, myocardial infarction, chronic kidney disease, and end-stage renal disease [2,3,4,5]. Hypertension results from complex interactions between genetic and environmental factors. Although the exact etiology of primary hypertension remains unclear, some risk factors (e.g., age, obesity, race, family history, reduced nephron number, high-salt diet, excessive alcohol consumption, and physical inactivity) are profoundly but independently associated with its development [6]. However, several epidemiological studies have indicated that high blood pressure arises from fetal life [7,8,9]—the so-called programming effect or the developmental origin of hypertension [10]. Paauw et al. [11] emphasized that pregnancy can be a window of opportunity to improve the long-term cardiovascular health of children; they proposed that improved perinatal care potentially reduces the incidence of hypertension and burden of cardiovascular disease in later life.

Iron-deficiency anemia (IDA) has become one of the most prevalent nutritional problems worldwide, particularly in preschool-age children, women of reproductive age, and even in high-income countries [12]. Pregnancy is associated with an increase in iron demand for the expansion of blood volume and growth of the fetus, placenta, and other maternal tissues, therefore, the risk of IDA is increased. Recent animal experiments have shown that prenatal IDA adversely affects offspring hypertension. The mechanism underlying the developmental disturbances in the pathological process of hypertension is poorly understood.

## 2. Results

### 2.1. Effect of Low-Iron Diet on Mother Rats

There was no significant difference in the dams’ body weight from 42 d/o to 63 d/o between the low-iron-diet and control-diet groups. Serum iron concentration was significantly lower in dams who had consumed the low-iron diet for 14 days than in those who had consumed the control diet. The mothers’ hemoglobin concentration was significantly lower after 3 weeks of the low-iron diet (Table 1).

### 2.2. Body Weight, Serum Iron, and Hemoglobin Concentration of Adult Male Offspring

At 16 weeks of age, the body weight of male offspring in the LLL group was lower than that of those in the other three groups. Serum iron concentration was lower in the LLL group than in the SC group. There was no significant difference in hemoglobin concentration among the four groups, which revealed that hemoglobin concentration recovered but serum iron concentration was still low after feeding with the control diet (Table 2).

SC, offspring of sham group. LCC, offspring of mother taking iron-deficient diet before mating. LLC, offspring of mother taking iron-deficient diet before mating and pregnancy. LLL, offspring of mother taking iron deficient diet before mating, pregnancy, and lactation.

### 2.3. Blood Pressure and Kidney Index

As shown in Figure 1, blood pressure was significantly higher in the LLL group than in the SC and LCC groups in the 8-week-old (w/o) offspring. Blood pressure was higher in the 16 w/o rats in the LLL group than in those in the other three groups. Kidney weight was lower in the LLC and LLL groups than in the SC group; however, there was no difference in kidney index (left kidney weight/body weight) among the four groups (Figure 2).

### 2.4. Renin Angiotensin System

The amount of angiotensin I (ANG I) in the renal cortex of the offspring was lower in the SC group than in the other three groups. The level of angiotensin-(1–7) (ANG 1–7) was lower in the LLL group than in the other three groups (Figure 3). As shown in Figure 4 and Figure 5, renal angiotensin II type 1 receptor (AT1R) and angiotensin-converting enzyme (ACE) abundance were higher in the LLL group than in the SC group. The abundance of angiotensin II type 2 receptor (AT2R) and MAS were lower in the LLL group than in the SC group. There was no difference in the abundance of ACE2 among the four groups.

### 2.5. Renal Cortex Sodium Transporter

The abundance of serum and glucocorticoid-inducible kinase 1 (SGK1) was lower in the LCC group than in the SC group. There was no difference in the level of sodium-chloride cotransporter (NCC) among the four groups (Figure 5).

### 2.6. Renal Cortex IL-6 and 8-OHdG

The abundance of interleukin 6 (IL-6) was higher in the LLC and LLL groups than in the SC and LCC groups (Figure 5). Higher 8-hydroxy-2′-deoxyguanosine (8-OHdG) intensities were observed in the LCC, LLC, and LLL groups than in the SC group (Figure 6).

## 3. Discussion

We reported that maternal iron deficiency may program adult male offspring blood pressure. Our study showed that the adult male 16 w/o offspring of a mother with a low-iron diet before pregnancy and throughout the lactation period had (1) higher blood pressure; (2) increased renal cortex AT1R and ACE abundance; (3) decreased renal cortex AT2R and MAS abundance; and (4) increased renal 8-OHdG and IL-6 abundance.

Barker et al. found that birth weight was inversely correlated with increased early death secondary to coronary heart disease [13]. Gluckman et al. proposed the concept of development of health and disease and emphasized the importance of influences on early development that interact with developmental plasticity to determine patterns of noncommunicable chronic diseases [14]. Gillman stated that it was implausible that a mother’s exposure to stress or toxins while pregnant, how she fed her offspring during infancy, and how fast the offspring grew during childhood can determine that offspring’s risk for chronic disease as an adult [15]. Therefore, maternal factors can influence a baby’s future profoundly.

Numerous factors are involved in the pathogenesis of the developmental programming of hypertension, including epigenetic processes, glucocorticoids, reduced nephron number, activation of the sympathetic nervous system and renin-angiotensin system (RAS), and endothelial dysfunction [16,17,18]. Suboptimal environmental conditions during fetal development, such as maternal illnesses, exposure to environmental chemicals, and medication use during pregnancy and lactation, are also reportedly relevant to the development of hypertension in adult offspring [19,20,21]. Maternal nutrition acts as a double-edged sword in the developmental programming of hypertension. An imbalance in maternal nutrition causes hypertension in offspring, and some nutritional interventions during pregnancy and lactation may work as reprogramming strategies to reverse programming processes and prevent the development of hypertension [22].

Recommendations for iron supplementation for pregnant women vary across countries, with no routine use by Canadian and Australian health authorities, but universal supplementation exists, with 30 mg/day of iron recommended by the United States Center for Disease Control and 30–60 mg/day recommended by the WHO [23,24]. In the UK, an iron supplement was suggested if the serum ferritin level was less than 30 ug/L [25]. In New Zealand, doctors screen for hemoglobin and serum ferritin levels of pregnant women at a gestational age of 26–28 weeks, and supply them with 65 mg/day if iron deficient and 130 mg/day if IDA [24]. In our study, we used the iron supplementation dose of 2.9 mg/kg in the experiment group and 52.3 mg/kg in the control group, which was published previously [26]. We can see the occurrence of lower serum iron concentration was before the drop of hemoglobin in dams who have consumed the low-iron diet. This mimics a common clinical scenario, in which some patients have IDA before pregnancy, when the increased demands or blood-volume expansion from pregnancy have not happened yet.

### 3.1. Programming Hypertension and IDA

Lewis et al. found that blood pressure was elevated in 3-month-old rats whose mothers were iron-restricted during pregnancy [27]. They also reported that systolic blood pressure was higher in the offspring of iron-restricted dams at 16 months of age [28]. Gambling et al. reported that male (but not female) pups born to iron-deficient dams had higher blood pressures than their normal counterparts [29]. In our study, we found that offspring of dams that consumed the low-iron diet before conception and throughout gestation and lactation had the highest blood pressure, which is comparable with the results of previous reports. The results of human studies were heterogeneous, however, some studies did not check for direct biomarkers of iron status, such as serum ferritin [30]. Lindberg et al. reported that low-birth-weight children who received iron supplementation (1 or 2 mg iron/kg/day) in infancy had lower systolic blood pressures at 7 years of age [31]. This observation suggests that the increased risk of hypertension observed in children and adults who are born smaller might be reduced with early iron supplementation [31]. It is worth noting that a significant decrease in serum iron levels occurs earlier than that in hemoglobin levels in dams fed a low-iron diet. Iron deficiency could be present before anemia and might cause programming effects despite normal hemoglobin levels. The speed at which iron deficiency is corrected can also affect the programming effect.

### 3.2. Programming Hypertension and RAS and IDA

The importance of the RAS in the developmental programming of hypertension has been shown by the ACE inhibitor to normalize blood pressure in undernourished offspring, relative to their control counterparts [32]. Vehaskari et al., reported that young offspring of low-protein dams showed decreased plasma renin activity and increased renal expression of AT1R, which was associated with an increase in plasma aldosterone levels [33,34]. In addition, the relative importance of RAS was demonstrated by the normalization of blood pressure observed with chronic RAS blockade [35].

Few studies have explored the role of the RAS in blood pressure programming by maternal dietary-iron restriction in rats. Lewis et al. reported that serum ACE concentrations were significantly elevated in the offspring of iron-restricted dams at 3 months but not at 14 months of age [28]. The elevation of blood pressure in iron-restricted offspring does not appear to be mediated by changes in ACE levels. Local regulation of the RAS and changes in other components of the RAS may contribute to the elevated blood pressure observed in this model. In our study, increased renal cortex AT1R and decreased levels of renal cortex AT2R, ACE2, and MAS reemphasized the fact that the RAS, whether classical or non-classical, played an important role in programming hypertension by IDA.

### 3.3. Programming Hypertension and Oxidative Stress and IDA

Oxidative stress (OS) is thought to be a cause, consequence, or potentiating factor in the development of hypertension [35]. Pretreatment of undernourished mothers during gestation with antioxidants prevents the development of hypertension in offspring [36]. Our previous work revealed that dimethyl fumarate administration during pregnancy protected adult offspring from the hypertension programmed by prenatal dexamethasone plus a postnatal high-fat diet (which are relevant to the downregulated mRNA expression of renin, angiotensinogen, ACE, and AT1R) [37]. In addition, maternal undernutrition is associated with increased OS in the placenta, indicating that exposure to OS originates in early life [38].

Woodman et al. reported that prenatal IDA caused hypoxia, mitochondrial dysfunction, and an increase in reactive oxygen species in term rat fetuses [39]. These adverse outcomes were organ- and sex-specific, and the kidneys of male fetuses exposed to prenatal ID were most affected [40], suggesting that renal development is particularly sensitive to programming effects. Aly et al. reported that dietary iron supplementation in pregnant women with IDA had an antioxidant effect with a significant decrease in the concentration of OS markers and an increase in antioxidant activity [41]. In our study, we found that increased renal cortex 8-OHdG intensity suggested that OS plays an important role in programming hypertension by IDA.

### 3.4. Programming Hypertension and Inflammation and IDA

Systemic inflammation contributes to the development of cardiovascular diseases, including endothelial dysfunction, atheroma plaque formation, acute thrombotic complications, and hypertension [42]. Maternal serum IL-6 concentrations are positively associated with fetal growth, thus linking the maternal proinflammatory environment to intrauterine overgrowth in obese mothers [43]. Maternal obesity and a high-fat diet reportedly program offspring hypertension [44]. Wei et al. reported that prenatal exposure to lipopolysaccharide resulted in increased blood pressure in rats [45]. In our study, we found that renal cortex IL-6 increased in the LLL group (which is compatible with previous reports), suggesting that inflammation plays an important role in programming hypertension.

### 3.5. Programming Hypertension and Sodium Transporter and IDA

Inappropriate tubular sodium reabsorption may play an important role in mediating hypertension (in adults) induced by adverse fetal environments. The upregulation of renal sodium transporters may contribute to hypertension in the fetal programming of hypertension. Bertram et al. found that the mRNA expression of the apical Na^+^-K^+^-ATPase pump increased in low-protein-exposed offspring [46]. Manning et al. reported that the thick, ascending-limb bumetanide-sensitive Na^+^-K^+^-2Cl^-^ cotransporter and distal convoluted tubule thiazide-sensitive NCC were upregulated prior to the development of hypertension [47]. In our study, we did not observe changes in SGK-1 and NCC in the LLL group, although hypertension was noted. Thus, the role of renal sodium transporters may differ in different stages and models of fetal programming of hypertension.

Our study has several limitations. First, we only studied male offspring in this study. Next, the relative markers in some possible mechanisms were only partially studied. Nevertheless, we have explored four possible mechanisms in programming hypertension. Further study, including female offspring in the experiments and a searching deprogramming strategy, is indicated for the future.

## 4. Materials and Methods

### 4.1. Animals and Experimental Design

We purchased 6-week-old virgin Sprague Dawley (SD) rats (BioLASCO Taiwan Co., Ltd., Taipei, Taiwan) and housed them in the animal care facility in Chang Gung Memorial Hospital, Kaohsiung, Taiwan in a 12 h light/dark cycle (the lights were turned on at 7 a.m.). All purified diets (Research Diets Inc.) were based on the AIN-93G diet and were similar in composition except for the iron concentration, which was 2.9 mg/kg in the low-iron diet (D03072501) and 52.3 mg/kg in the control diet (D10012G).

Blood was drawn from the tail vein of SD female rats, and the iron profile and hemoglobin concentrations were checked every week for 3 weeks, after feeding them with their respective diets. Next, they were allowed to mate with the male rats for 24 h. They were then separated from the male rats and housed individually in a standard plastic home cage, one day later. Dams consuming the control diet were maintained on it throughout pregnancy and the lactation period. Dams consuming the low-iron diet were randomly divided into three groups: (1) control diet through pregnancy and lactation; (2) low-iron diet during pregnancy, but control diet during the lactation period; and (3) low-iron diet throughout pregnancy and lactation. To decrease gender interference, only male rats were used in this study. Their offspring were, therefore, categorized into four groups according to maternal diet: (1) SC group: offspring of dams fed the control diet throughout the whole course; (2) LCC group: offspring of dams fed a low-iron diet before pregnancy; (3) LLC group: offspring of dams fed a low-iron diet before lactation; and (4) LLL group: offspring of dams continuously receiving the low-iron diet. All the offspring received the control diet after weaning; they were sacrificed at 16 + 1 weeks of age by rompun + zoletil and exsanguination (Appendix A Appendix A). Blood and kidney samples were collected for further analysis. The protocols described herein were approved by the Animal Care and Use Committee (Chang Gung Memorial Hospital, Kaohsiung, Taiwan, No. 2019030502), with minimal animal suffering during the experiments.

### 4.2. Blood Pressure Measurement

We measured the blood pressure of conscious rats at 8 and 16 weeks of age, using the indirect tail-cuff method (BP-2000, Visitech Systems, Inc., Apex, NC, USA) after systematic training. To ensure accuracy and reproducibility, the rats were adapted to restraint and tail-cuff inflation for 7 days before the experiment commenced, and measurements were taken between 1:00 PM and 5:00 PM daily. Rats were placed on the device platform and their tails were passed through tail cuffs and secured in place with a tape. After a 10-min warm up period, 10 preliminary cycles were performed to allow the rats to adjust to cuff inflation. For each rat, five measurements were recorded at each time point, as previously described [48]. Three consecutive stable measurements were obtained and averaged.

### 4.3. Immunohistochemistry

Reactive oxygen species can damage cellular structures such as nucleic acids, proteins, and lipids. Hydroxyl radicals damage all components of DNA molecules, including purine, pyrimidine, and deoxyribose structures; 8-OHdG is an important biomarker for the evaluation of oxidative DNA damage [49]. For immunohistochemical staining, the renal cortex was dissected, fixed, cut into sections (4 μm thick), and transferred onto polylysine-coated slides. Sections were immunostained for 8-OHdG. Immunoreactivity was demonstrated using horseradish peroxidase-3′-diaminobenzidine cell- and tissue-staining kits.

### 4.4. Enzyme-Linked Immunosorbent Assay (ELISA)

The RAS plays an important role in the regulation of renal, cardiac, and vascular physiology, and its activation is central to many common pathologic conditions, including hypertension. Plasma ANG I and ANG 1-7 levels were determined by ELISA with adequate dilution, as indicated.

### 4.5. Western Blot Assay

Infiltration of innate and adaptive immune cells and other inflammatory processes in the kidneys and other organs occurs in individuals with hypertension [50,51]. Of these cells, IL-6 is positively correlated with hypertension [52]. The expression and phosphorylation of the NCC are regulated by dietary salt, potassium, and SGK1 and affect blood pressure regulation [53]. Renal AT1R, AT2R, ACE, ACE2, MAS, SGK 1, NCC, and IL-6 were examined by Western blotting, as previously described [54]. Briefly, measurements were conducted on the kidney (100–200 μg of total protein). We used primary antibodies, including AT1R, AT2R, ACE, ACE2, MAS, SGK 1, and IL-6, followed by secondary antibodies. Bands of interest were visualized using enhanced chemiluminescence reagents and quantified by densitometry as integrated optical density (IOD) after subtracting the background. The IOD was factored for Ponceau S staining to correct for any variations in total protein loading. Protein abundance was represented as IOD/PonS.

### 4.6. Statistical Analyses

Biochemical parameters, pathology, and Western blot data were analyzed by one-way analysis of variance with a least significant difference post hoc test. All analyses were performed using SPSS software on a compatible personal computer. Values were expressed as mean ± standard error of mean, and significance was defined as *p* < 0.05 for all tests.

## 5. Conclusions

This study is likely to enhance our understanding of the mechanisms by which maternal iron-deficiency affects blood pressure in offspring. Improving the iron status of pregnant mothers could positively influence the future development of hypertension in their offspring. These studies have major potential clinical significance because many adult diseases may develop in children, even before birth.

## Figures and Tables

**Figure 1 ijms-23-08294-f001:**
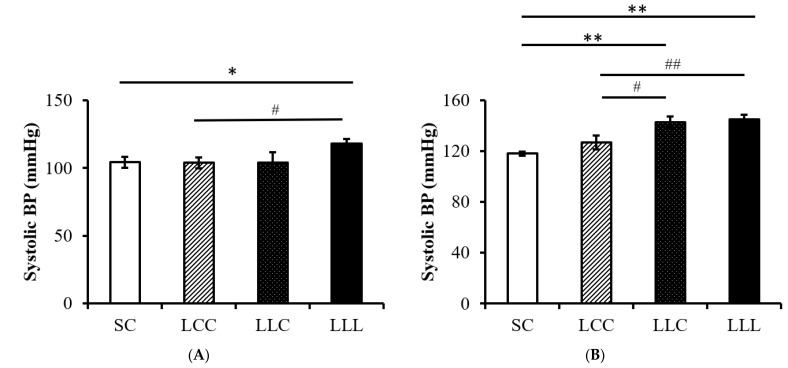
Blood pressure of offspring. (**A**) At 8 weeks old (N = 8 in each group). (**B**) At 16 weeks old. Four groups were evaluated by one-way ANOVA with the LSD post hoc test. * vs. SC, *p* < 0.05; ** vs. SC, *p* < 0.01; # vs. LCC, *p* < 0.05; ## vs. LCC, *p* < 0.01.

**Figure 2 ijms-23-08294-f002:**
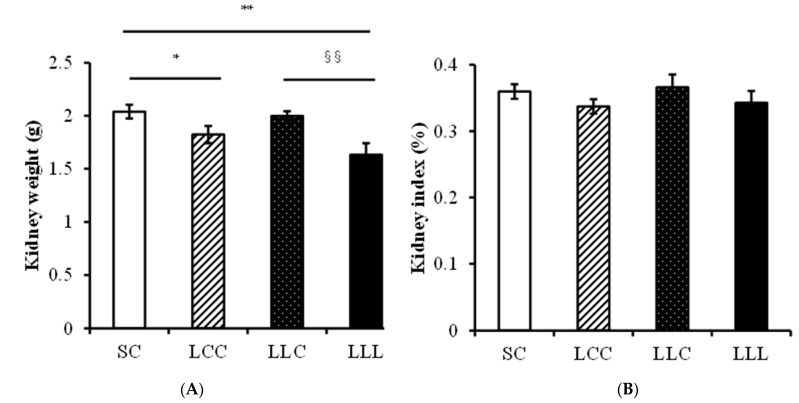
Kidney weight and index. (**A**) The left kidney weight (**B**) The kidney index. Four groups were evaluated by one-way ANOVA with the LSD post hoc test. * vs. SC, *p* < 0.05; ** vs. SC, *p* < 0.01; §§ vs. LLC, *p* < 0.01.

**Figure 3 ijms-23-08294-f003:**
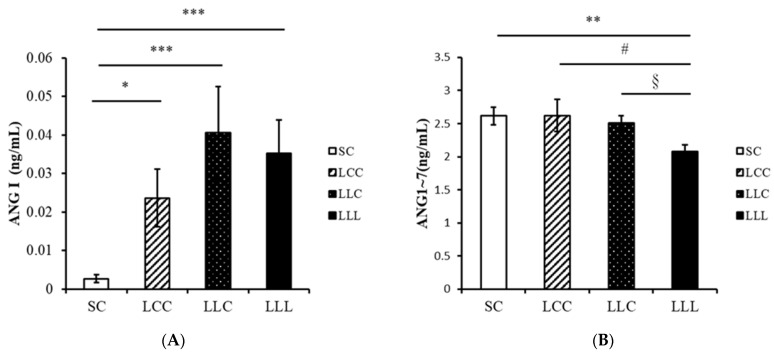
Offspring renal cortex ANG I (**A**) and ANG 1-7 (**B**) levels. Four groups were evaluated by one-way ANOVA with the LSD post hoc test. * vs. SC, *p* < 0.05; ** vs. SC, *p* < 0.01; *** vs. SC, *p* < 0.001; # vs. LCC, *p* < 0.05; § vs. LLC, *p* < 0.05.

**Figure 4 ijms-23-08294-f004:**
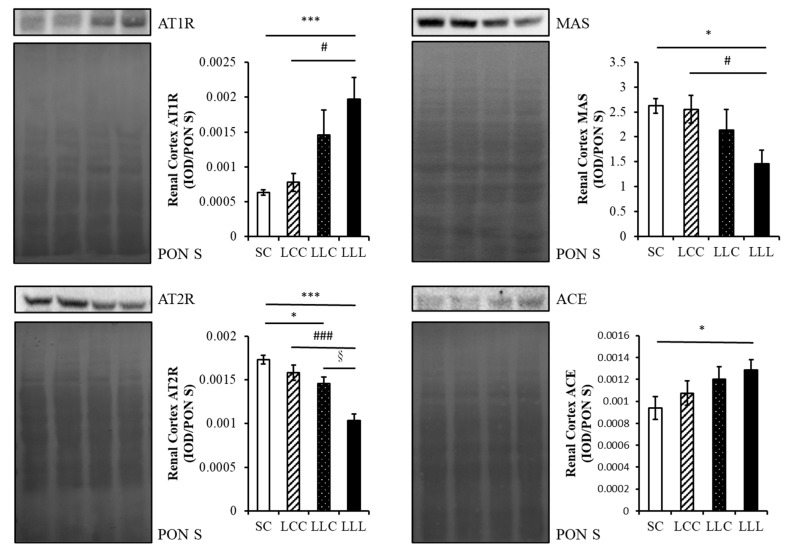
Offspring renal cortex AT1R, AT2R, MAS, and ACE abundance. Four groups were evaluated by one-way ANOVA with the LSD post hoc test. * vs. SC, *p* < 0.05; *** vs. SC, *p* < 0.001; # vs. LCC, *p* < 0.05; ### vs. LCC, *p* < 0.001; § vs. LLC, *p* < 0.05.

**Figure 5 ijms-23-08294-f005:**
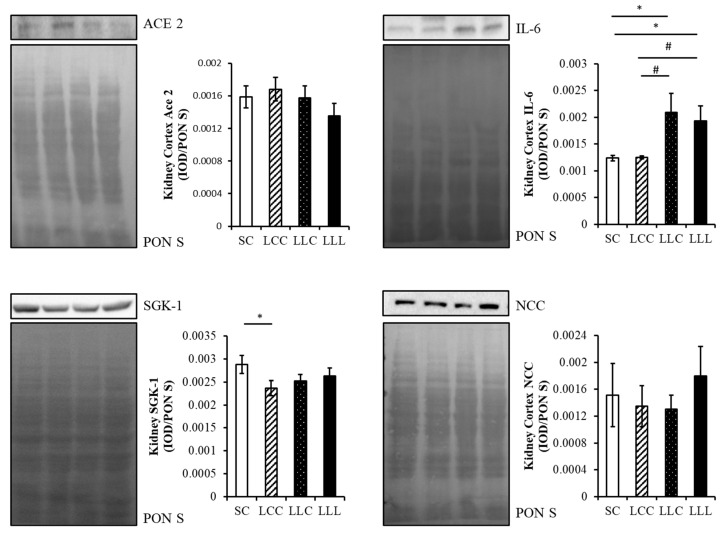
Offspring renal cortex ACE2, IL-6, SGK1, and NCC abundance. Four groups were evaluated by one-way ANOVA with the LSD post hoc test. * vs. SC, *p* < 0.05; # vs. LCC, *p* < 0.05.

**Figure 6 ijms-23-08294-f006:**
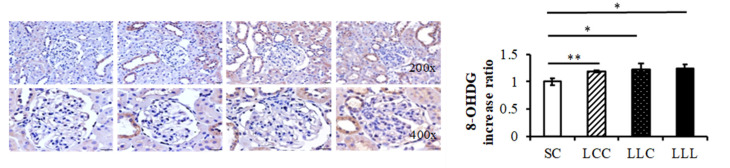
Offspring renal cortex 8-OHdG stain. Four groups were evaluated by one-way ANOVA with the LSD post hoc test. * vs. SC, *p* < 0.05; ** vs. SC, *p* < 0.01.

**Table 1 ijms-23-08294-t001:** Mothers’ body weights, serum iron, and hemoglobin concentrations after taking control or low-iron diet (from 42 d/o) (N = 6 in each group).

Day-Old		42	49	56	63
**Body weight (g)**	Control diet	180.5 ± 16.9	200.5 ± 13.5	221.9 ± 16.9	239.8 ± 14.1
	Low-iron diet	184 ± 10	204 ± 7.2	228.5 ± 10	243.7 ± 13.2
**Serum Iron (μM)**	Control diet	58 ± 7.4	58.2 ± 2.7	70.1 ± 9.7	42.6 ± 5.9
	Low-iron diet	38.3 ± 1.2	40.5 ± 0.6	23.9 ± 4.2 **	16.4 ± 4.5 **
**Hemoglobin (g/dL)**	Control diet	14.5 ± 0.1	14.7 ± 0.4	14.4 ± 0.5	14.8 ± 0.1
	Low-iron diet	14.2 ± 0.1	14.6 ± 0.3	13.3 ± 0.5	12± 0.6 *

* vs. control diet, *p* < 0.05; ** vs. control diet, *p* < 0.01.

**Table 2 ijms-23-08294-t002:** Adult male offspring’s body weights, serum iron, and hemoglobin concentrations at 16 w/o (N = 8 in each group).

Group	SC	LCC	LLC	LLL
	Mean ± SD			
**Body weight (g)**	567.1 ± 6.2	542.1 ± 24.8	552.4 ± 24.7	463.75 ± 13.3 ** ^#^ ^§^
**Serum Iron (** **μM)**	33.5 ± 1.5	28.5 ± 5.8	22.6 ± 5.2	19.6 ± 4.2 *
**Hemoglobin (g/dL)**	15.9 ± 0.3	16 ± 0.2	15.9 ± 0.5	15.5 ± 0.5

* vs. SC, *p* < 0.05; ** vs. SC, *p* < 0.01; ^#^ vs. LCC, *p* < 0.05; ^§^ vs. LLC, *p* < 0.05.

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
