# Peer review of "Maternal Iron Deficiency Programs Rat Offspring Hypertension in Relation to Renin—Angiotensin System and Oxidative Stress"

_ijms, 2022, doi:10.3390/ijms23158294_

Round 1
Reviewer 1 Report
The present study demostrated the importance of micronutrients during early periods of life for programming cardiovascular and renal alterations in adult life.
however there are some queries:
1- The animal protocol is not clear enough. I recommend the authors to introduce a scheme with the mother groups, offspring groups, periods of iron deficiency and age of mothers and offsprings.
2- In table 1, it is not clear the time periods when mothers were studied previos mating with males. This could be better understand if the animal protocol is better explained.
3-How many male offspring remained with their mothers after birth?
4-Considering the relationship between blood pressure and birth weight, have you determined birth weight?
5-Define kidney index
6-Have the authors determine urine volume, sodium excretion, glomerular filtration rate? to correlate with changes in renal renin angiotensin sytem?
Author Response
Thank you very much for your letter. We greatly thank for your kindness to give us opportunity for revision.
Please find below a point-by-point response to reviewers’ comments

Reviewer 2 Report
The authors aimed to study changes in blood pressure, the renal renin-angiotensin system, oxidative stress, inflammation, and sodium transporters in adult male offspring of maternal low-iron diet Sprague-Dawley rats. The results showed that 16- week-old male offspring born to mothers with low dietary iron throughout pregnancy and the lactation period had higher blood pressure, increased renal cortex angiotensin II receptor type 1 and angiotensin-converting enzyme abundance, decreased renal cortex angiotensin II receptor type 2 and MAS abundance, and increased renal 8-hydroxy-2'-deoxyguanosine and Interleukin-6 abundance. They concluded that improving the iron status of pregnant mothers could influence the development of hypertension in their offspring.
This is a well-designed and nicely presented study.
Comments:
· Based on the title it is not obvious that it is an animal study. Therefore, the title should be modified.
· Why these doses of iron supplementation (2.9 mg/kg vs. 52.3 mg/kg) were used? Is the degree of anemia being similar in rats compared to the degree of anemia in human iron deficiency?
· The current guidelines regarding iron supplementation in pregnancy should be discussed and cited.
· Figure 1B: title of y-axis should be corrected
· Ref 11. should be checked (2017219:241-59.).
Author Response
Thank you very much for your letter. We greatly thank for your kindness to give us opportunity for revision.
Please find below a point-by-point response to reviewers’ comments:

Round 2
Reviewer 1 Report
The manuscritp was improved. The authors should consider the mantain the same number of offspring after birth to ensure that an adequate y similir feeding during lactation, considering tha rats develop 12 breasts.